# Spiderweb Cellular Structures Manufactured via Additive Layer Manufacturing for Aerospace Application

**Klaudio Bari \*** and **Lucie Bollenbach**

School of Engineering, Telford Innovation Campus, University of Wolverhampton, Telford TF2 9NN, UK; l.bollenbach@wlv.ac.uk

\* Correspondence: K.Bari@wlv.ac.uk; Tel.: +44-1902-32-3845

**Abstract:** With increasing the energy costs and aiming for fossil-free Europe, cellular structures could provide a cost-effective tool for saving fuel consumption in aircraft. To achieve this goal, a cellular structure topology is a rapidly growing area of research facilitated by developments in additive layer manufacturing. These low-density structures are particularly promising for their aerospace applications. In this paper, four cellular structure topologies are developed to serve as a vibration damper in small electric aircraft motor, we have compared their performance with the original motor holder in the aircraft. This paper introduces the roadmap of scaffolding concept design and provides a novel concept in vibration damping. Based on the FEA simulation, aluminium 6061T spiderweb-inspired lattices (weight 0.3473 g and porosity 84%) have proven to have the lowest natural resonance and highest yield strength to weight ratio compared to other scaffolding concepts.

**Keywords:** cellular materials; vibration damper; Ansys model simulation; brushless motor; spider web; snowflake unit cell

## 1. Introduction

In the field of material science, there have been many exciting recent developments providing new possibilities for engineering solutions, but arguably the simplest and most practical of these advancements is that of cellular materials. Cellular materials apply the same structural principles of large-scale structures to the mesoscale, creating materials acting as vibration dampers [1]. One common problem in single propulsion motors in electric aircraft is the high spinning speed of the propeller to gain thrust, hence enormous random vibration is generated. Figure 1 shows a streamliner 350 cm wingspan, it is autonomous electric aircraft that is used for border surveillance purposes in the Shropshire region in the United Kingdom. One of the technical tasks given by the University of Wolverhampton is to redesign the front motor holder to mitigate vibration to save power and increase the range of the flight.

In the past, various concepts were designed and manufactured by a group of ESTIA students in the frame of their assessment in the Emerging Design Tool module (7AT004) at the University of Wolverhampton, as shown in Figure 2. Some of the studies compared the performance of various aluminium/titanium alloys [2], and others explored the solver computation limit of Ansys software to achieve the most accurate outcomes [3].

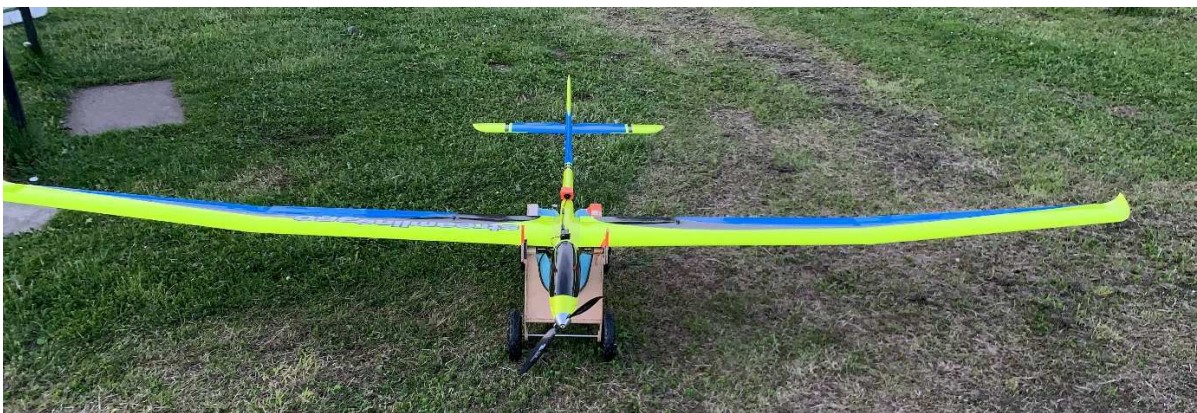

**Figure 1.** The Streamliner 350 aircraft single "16 × 10" propeller at the front of the fuselage.

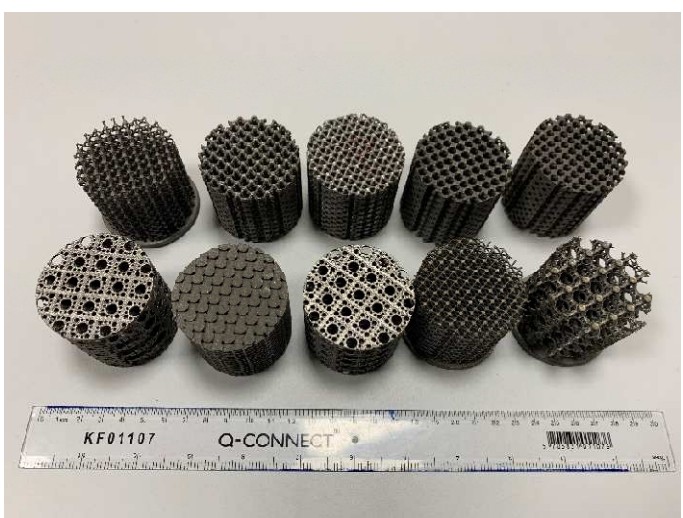

**Figure 2.** Various cellular concept cubes made from Ti-64 alloy.

As the density of the cellular material decreases, so do its other properties, proportionally so, or more than proportionally. Specific strength and stiffness cannot be increased from that of the base material. In applications where low weight is required, but a certain volume or surface must be filled, cellular structures are ideal [4]. Materials with high specific strength and stiffness, as well as other desirable properties, can be used and their density decreased as much as needed in applications when they would normally have been too heavy. This study focuses on the design and manufacturing of super lightweight motor holders. The process usually starts from a unit cell and then a full scaffolding structure to obtain the highest strength to the ratio [4]. Advances in additive manufacturing and lightweight high strength materials have allowed more unusual and tailor-made aircraft parts to be constructed. Ultimately this will reduce noise and electric power consumption in the aircraft and elongate the flight range [5]. The scope of this study is specifically aimed at a one-stop concept to help designers establish a roadmap for developing cellular structure components [6]. There are many deliverables anticipated for this project, not the least of which is the analysis of two new and innovative spider web cellular designs in comparison to two traditional cellular architectures. Comparisons are a common theme in this study as a unit cell (UC) and full-scale cellular simulations are comparatively assessed for their merits, for which there has not been enough research. Simulations will then be compared to real-world testing to assess their validity and accuracy. The robustness of the mechanical idealisms will also be tested by way of the disparity between the intended material stiffness and the true stiffness. However, further research is needed to assess material performance in more detail. This study will only consist of simple axial compression testing, if this

is successful, then designs can proceed to the next stages, which are fatigue testing and low-speed impact testing [6]. Materials that are designed on the micro or mesoscale to contain pores are cellular materials. These can be classically categorized as foams, honeycombs, and lattices. Another important classification is the regularity of cavities within the object, stochastic or periodic [7]. Stochastic closed porosity is naturally occurring and anisotropic, seen in a lot of materials on the microscale and on the macroscale in foams. Uniform periodic cellular materials are formed of equal repeatable cavities. Hierarchical cellular materials, however, are a special class of periodic materials as their cavities vary in the shape of size throughout the material according to a predefined pattern [8]. More technically, one can organize these materials by the number of open directions in cartesian space, as shown in Table 1.

**Table 1.** Classification of cellular material topologies by open direction and uniformity.

| Open Directions | Examples | |
|---|---|---|
| | **Stochastic** | **Periodic** |
| 0, Fully Closed | Foam | Pocketed solid |
| 1, Prismatic Open | Wood (tracheae) | Honeycomb-based material  Corrugated panelling |
| 2, Planar Open | | Multi-layered, strut-supported sandwich structure |
| 3, Fully Open | Natural spongeCancellous bone | Lattice structure |

This topology categorisation works best, in terms of Euclidean geometry, for regular third-order polyhedral, cubic-based shapes. Topology categorisation is specific to the internal geometry of the cellular material being used [7]. In this project, the materials being designed will be open periodic cellular (lattice) materials, though future work should expand into hierarchical materials [9]. A comparison between the different types of cellular materials in terms of the weight, damping factor, and mechanical properties will be carried out. Prismatic topologies, such as honeycombs and corrugated panelling, were mainly investigated in this paper. Hierarchical cellular materials can offer significant improvements, when designed correctly, to uniform periodic materials [8]. On this scale level, the stresses, base material micro-porosity, surface roughness, surface area, and other micro-mechanical properties may be assessed. Cellular materials can also, however, be viewed as bulk materials, measuring their macro-mechanical behaviour, such as bulk strain, bulk strength, and bulk deformation behaviour. Various measurement parameters can be used to define a lattice material relative to the base material used to form it. Of these, the most influential factor is the relative density of the lattice material, which is also the volume fraction of solid material and the inverse of the bulk porosity [7]. The basic equation for the relative density of cellular material is shown in Equation (1).

$$Lattice\ Material\ Relative\ Density,\ \rho_r = \frac{\rho_l}{\rho_b} = \frac{V_b}{V_l} = \frac{1}{p_l} = \frac{V_l}{V_p} \tag{1}$$

where; $\rho_l$ = Lattice Bulk Density
$\rho_b$ = Base Material Desnity
$V_b$ = Volume of the Base Material Used in Lattice
$V_l$ = Volume of the Bulk Lattice Material
$p_l$ = Lattice Bulk Porosity
$V_p$ = Volume of Pores Within Bulk Material

## 2. Methodology

For periodic cellular materials, the entire structure can be defined by a representative volume element that contains the smallest linearly repeatable geometry, called a UC. By multiplying this UC out, a cellular material of any scale can be made. It, therefore, stands that, as the relative density will remain unchanged, the mechanical properties of a lattice

are unaffected by the scale or number of UC used to make it. The deformation behaviour of the material and material stress distribution are hence often approximated from that of a single UC [8].

Typically, a lattice structure will be modelled as a bulk material with homogenised properties, and a single unit cell will be assessed in FEA to validate the homogenised approach. One of the purposes of this project is to assess the differences in FEA between the UC model and lattices comprised of multiple unit cells. Throughout this project, one of the primary material properties of interest is stiffness, however, the distinction must be made between stiffness and Young's modulus as these terms are often interchanged, see Equation (2).

$$Stiffness, \; k = \frac{A \cdot E}{L} \tag{2}$$

where:

$k = material\ stiffness$
$A = cross - sectional\ area,$
$E = Young's\ modulus\ (or\ elastic\ modulus) = \frac{stress}{strain}$
$L = length\ in\ direction\ of\ deflection$

The design of the unit cell is a very important first step in the design concept since it will be formed by scaffolding unit cells. Different designs were, therefore, tested, but not all of them could be selected for further evaluation either because the unit cell did not meet the required strength or stiffness or because of the limited capacity of the computational solver. However, some of them are still interesting and promising, which is why they are presented in this paper.

The first idea was to take inspiration from acoustic foam used to absorb acoustic waves, thinking that if this form absorbs acoustic waves, it can be effective for vibration damping [1]. Two shapes were designated with different dimensions: One with a height of 8 mm and the other with a height of 4 mm (Figures 3 and 4). Unfortunately, this shape has been abandoned because of its mesh complexity. The other three-cell units (Figures 5–7) were used to build the final scaffolding part.

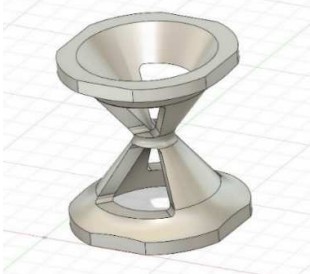

**Figure 3.** Acoustic foam inspired unit cell ($4 \times 4 \times 8$ mm$^3$).

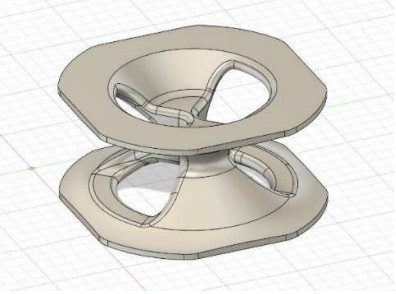

**Figure 4.** Acoustic foam inspired unit cell ($8 \times 8 \times 4$ mm$^3$).

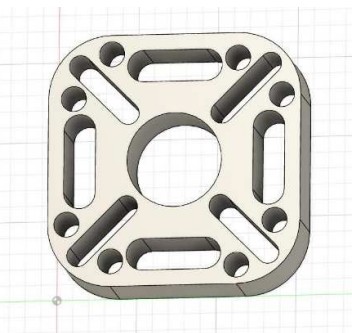

**Figure 5.** Polyhedral unit cell (8 × 8 × 2 mm³).

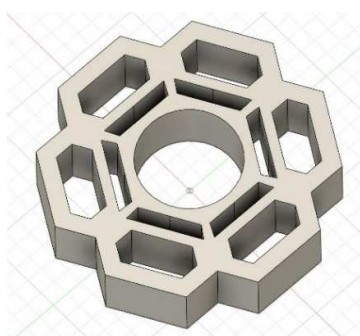

**Figure 6.** Snowflake inspired unit cell (8 × 8 × 2 mm).

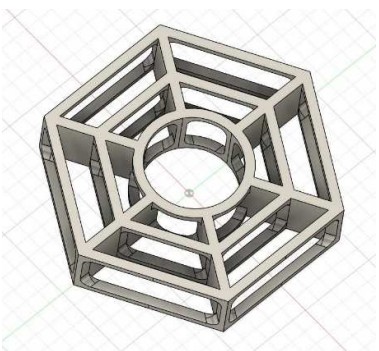

**Figure 7.** Spider web-inspired unit cell (8 × 8 × 2 mm³).

Table 2 lists the unit cells' area, mass, and volume to compare their densities. Four motor holders were made using polyhedral, snowflake, circular, and linear spiderwebs, as shown in Figure 8a–d, respectively.

**Table 2.** Unit cells properties.

|  | Polyhedral Unit Cell | Snowflake Unit Cell | Spider Unit Cell |
|---|---|---|---|
| Area (mm²) | 287.036 | 251.299 | 184.554 |
| Mass (g) | 0.166 | 0.132 | 0.047 |
| Volume (mm³) | 61.463 | 49.008 | 17.407 |

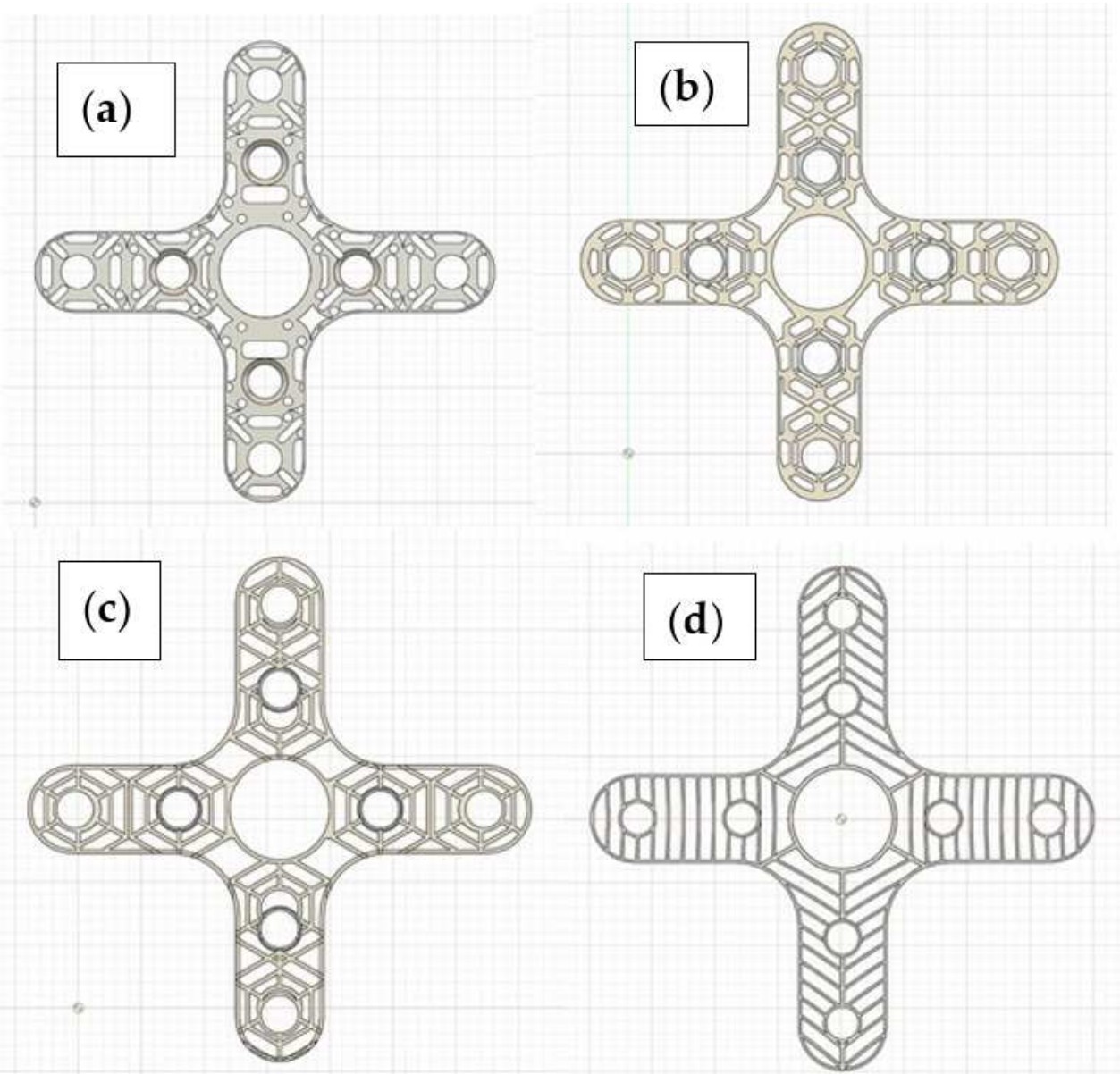

**Figure 8.** Final lattice structure to be used as a motor holder, (**a**) polyhedral, (**b**) snowflake, (**c**) circular spider web, (**d**) linear spider web.

The original motor holder shown in Figure 9a is currently in use in the aircraft to bolt thrust motor to the fuselage. It is solid aluminium composed of basic four holes to bolt the motor and other four holes to attach the motor to the fuselage, as shown in Figure 9b.

The porosity of each holder shown in Figure 8 is calculated using the following formula:

$$P = \left( 1 - \frac{The\ volume\ of\ solid\ material}{Total\ volume} \right) * 100$$

The volume of the holder corresponds to that of the full holder: $V_{holder} = 803.995$ mm$^3$

The different properties of the holders compared to the original holder are listed in Table 3.

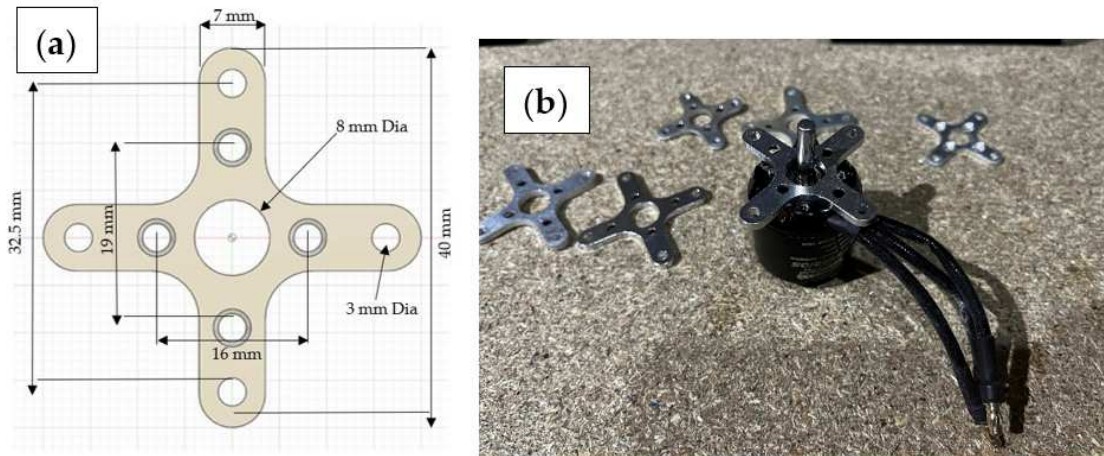

**Figure 9.** Original motor holder for streamliner 350 autonomous aircraft (**a**) original CAD model of the holder, (**b**) physical motor assembly with the holder.

**Table 3.** Holder's physical properties.

|  | Polyhedral | Honeycomb | Circular Web | Linear Web | Original |
|---|---|---|---|---|---|
| Area (mm$^2$) | 2094.02 | 2199.422 | 2017.546 | 2000 | 1279.349 |
| Mass (g) | 1.324 | 1.123 | 0.6132 | 0.3473 | 2.171 |
| Mass in comparison to the full holder (%) | 61 | 52 | 28 | 16 | 100 |
| Volume (mm$^3$) | 490.531 | 415.747 | 227.112 | 128.621 | 803.995 |
| Porosity (%) | 39 | 48.3 | 71.8 | 84 | 0 |

Having high porosity in such a structure means the weight has been greatly reduced. The linear spider web holder shows high porosity compared to other counterparts. However, a numerical analysis is needed to ensure that the structure will have enough structural integrity during flight.

## 3. Results

Two different simulations will be carried out for each holder. First, static analysis will allow us to determine the von Mises criterion, such as safety factors and stress. Then, a dynamic analysis to obtain the different vibration modes of the supports. Identical boundary conditions were applied to the different holders for both the static and dynamic analysis, and they have all the same thickness of 2 mm.

Steps in conducting the simulations:

(1) Apply the right material to the support, i.e., 6061 aluminium.
(2) Apply the force of gravity in the right direction with a value of 9.807 m/s$^2$
(3) Apply the different forces to the four holes closest to the central hole:

  − the one corresponding to the weight of the motor
  − the one corresponding to the tensile force of the propeller

(4) Fix the support at its ends
(5) For the meshing: Apply an absolute size of 1 mm curvature element order
(6) Generate mesh
(7) Pre-check the study setup
(8) Solve

The calculation for the force value of the motor weight:

$$\vec{P} = m * \vec{g}$$

Brushless motor mass = 200 g = 0.2 kg

$$\vec{g} = 9.81 \text{ m.s}^{-2} \approx 10 \text{ N kg}^{-1}$$

$$\vec{P} = 0.2 * 10 = 2 \text{ N}$$

Weight of brushless motor = 2 N

In order to calculate the propeller tensile force, we first had to calculate the static generated thrust using the data in Table 4.

**Table 4.** Values used to calculate the static thrust.

| Data | Values |
|---|---|
| Standard propeller | 16″ × 10″ inch |
| Number of blades | 2 |
| Rotations per minute (RPM) | 10,000 RPM |
| Motor weight | 100 g |
| Air Density | 1.1648 km m³ |
| Length of the motor | 50 mm |

The value of the static thrust obtained is 1.19 kg

$$\vec{P} = m * \vec{g} = 1.19 * 9.81 \approx 11.67 \text{ N}$$

Tensile force of the propeller = 11.67 N

Ansys software was used for simulations of both unit cells and holder using curvature mesh and bespoke boundary conditions illustrated in Figure 10, such as flat fixtures, remote weight (100 g), and tensile force 11.67 N for a static test. For dynamic test using spin rotor solver module in Ansys software, the boundary conditions in Table 4 were used. It is important to mention that the thickness of the holder was not changed, it was kept at 2 mm as specified by the manufacturer of streamliner D-power in Germany.

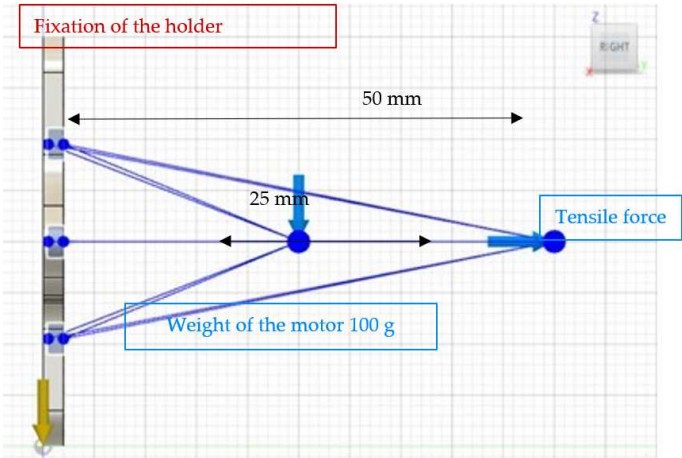

**Figure 10.** Parameters of the simulations.

### 3.1. Unit Cells Simulations

A modal frequencies simulation of the unit cells was carried out before designing the holders to check that they had the necessary characteristics. Only the gravity of $-9.807 \text{ m/s}^2$ was applied, and one side of the cell unit was fixed. When forecasting possible failure modes or the types of analysis required to fully understand performance, it is imperative to understand natural frequency. Each design has preferred frequencies of vibration, resonant frequencies, which are characterised by specific shapes of vibration. In our frequency analysis, we used an Eigenvalue approach to establish the natural modes of vibration for all three unit cells shown in Figure 11a–c. When a design's natural modes and its expected surface vibration environment are too closely matched, it can lead to a harmonic resonance, which can result in excessive loads causing failure.

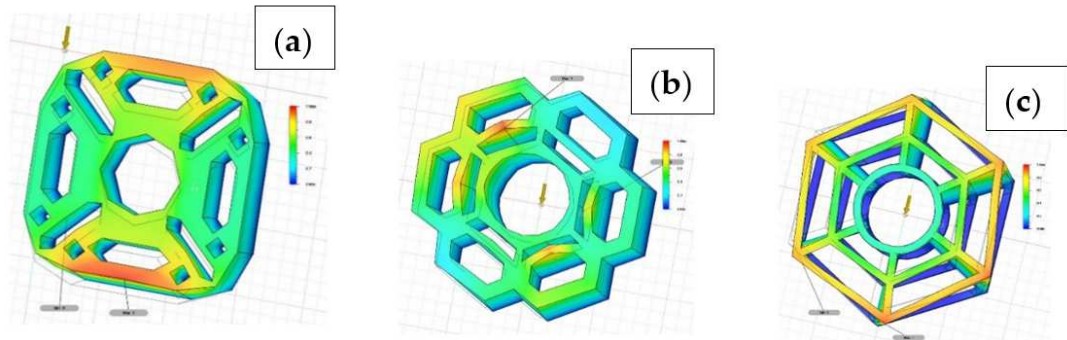

**Figure 11.** Modal frequencies simulations of the unit cell, (**a**) polyhedral unit cell: non homogenous top/bottom resonance accumulation, (**b**) snowflake, bipolar resonance accumulation (**c**) spiderweb unit cell: perfect uniform resonance accumulation..

The results of the modal frequency simulation of those three unit cells are listed in Table 5

**Table 5.** Modal frequencies simulation results.

|  | Polyhedral | Honeycomb | Spider Web |
|---|---|---|---|
| Mode 1 | 318,586 Hz | 289,734 Hz | 137,224 Hz |
| Mode 2 | 319,067 Hz | 290,432 Hz | 172,058 Hz |
| Mode 3 | 341,272 Hz | 293,151 Hz | 173,788 Hz |
| Mode 4 | 362,719 Hz | 294,774 Hz | 187,356 Hz |
| Mode 5 | 371,649 Hz | 330,268 Hz | 187,828 Hz |
| Mode 6 | 402,251 Hz | 335,592 Hz | 193,021 Hz |
| Mode 7 | 419,680 Hz | 362,079 Hz | 197,761 Hz |
| Mode 8 | 421,437 Hz | 369,414 Hz | 199,929 Hz |

Spider web shows the lowest natural frequency in all dynamics eight modes compared to other competitors, hence, the best as vibration damper and less noise generation during flight

### 3.2. Holders' Simulations

FEA simulations using static stress analysis were performed using Ansys software by applying the boundary condition illustrated in Figure 10 on those four holders. The stress static results of von Mises stress are presented in Figure 12a–e.

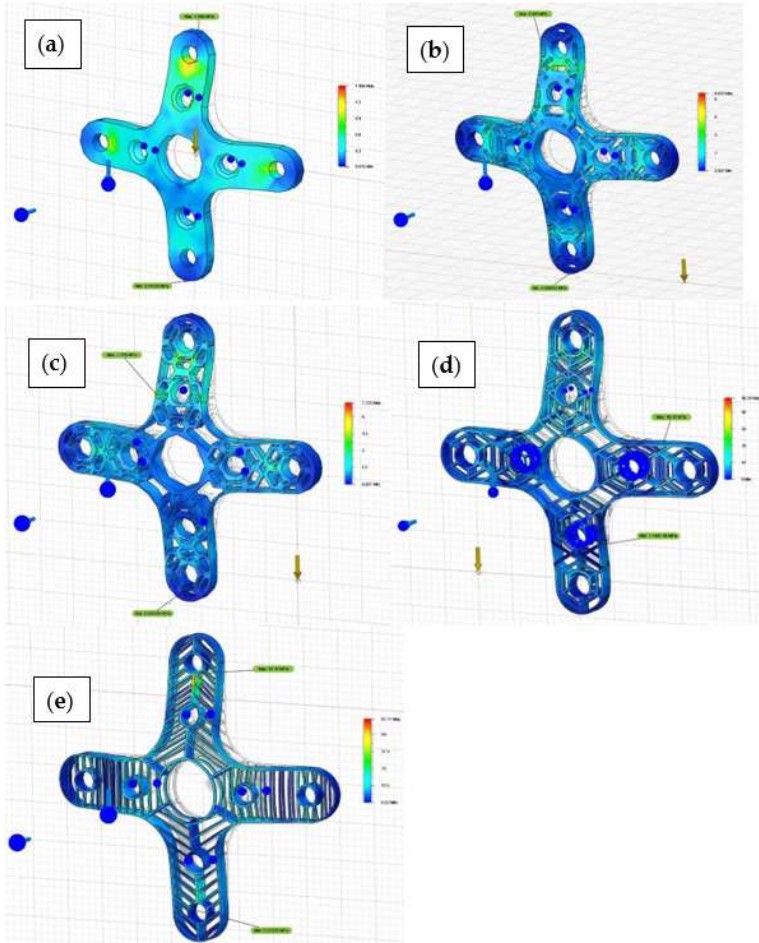

**Figure 12.** FEA static stress analysis of (**a**) original holder, (**b**) polyhedral, (**c**) snowflake, (**d**) circular spiderweb, and (**e**) linear spiderweb.

The von Mises criterion is verified using the stress and safety factor. It must be less than the Yield Strength of the material, Aluminium 6061, i.e., 275 MPa. All holders were in a safe range of elasticity limit of Aluminum 6061. Table 6 summarizes the outcomes of the stress analysis.

**Table 6.** Static stress simulation results.

| Holder Original | Minimum | Maximum |
|---|---|---|
| Stress (MPa) | $1.53 \times 10^{-2}$ | 1.498 |
| Safety factor | >15 | >15 |
| Displacement (mm) | 0 | $5.13 \times 10^{4}$ |
| Holder polyhedral | Minimum | Maximum |
| Stress (MPa) | 0.006613 | 8.643 |
| Safety factor | >15 | >15 |
| Displacement (mm) | 0 | 0.001871 |
| Holder honeycomb | Minimum | Maximum |
| **Stress (MPa)** | 0.001244 | 7.235 |
| **Safety factor** | >15 | >15 |
| **Displacement (mm)** | 0 | 0.001241 |
| **Holder circular Spiders web** | Minimum | Maximum |
| **Stress (MPa)** | $1.75 \times 10^{-6}$ | 48.34 |
| **Safety factor** | >15 | 5.73 |
| **Displacement (mm)** | 0 | 0.009074 |
| **Spider web linear holder** | Minimum | Maximum |
| **Stress (MPa)** | 0.02376 | 61.11 |
| **Safety factor** | >15 | 4.5 |
| **Displacement (mm)** | 0 | 0.02073 |

The von Mises stress and displacement criterion are checked for each holder. They must be in line with manufacturer D-power guidance not to exceed 0.1 mm during flight. This small-displacement condition means that the fuselage can handle such deflection because it is made from carbon fibre reinforced polymer [9].

In fields such as aeronautics and aerospace, it is important to have a high safety factor. The design of the parts must not compromise this parameter, it must, above all, be safe [1]. For any holder, a minimum safety factor of 3 is required.

### 3.3. Modal Frequencies Analysis

The modal frequencies analysis is carried out in the same way as static stress analysis. The results are presented in Figure 13a–e.

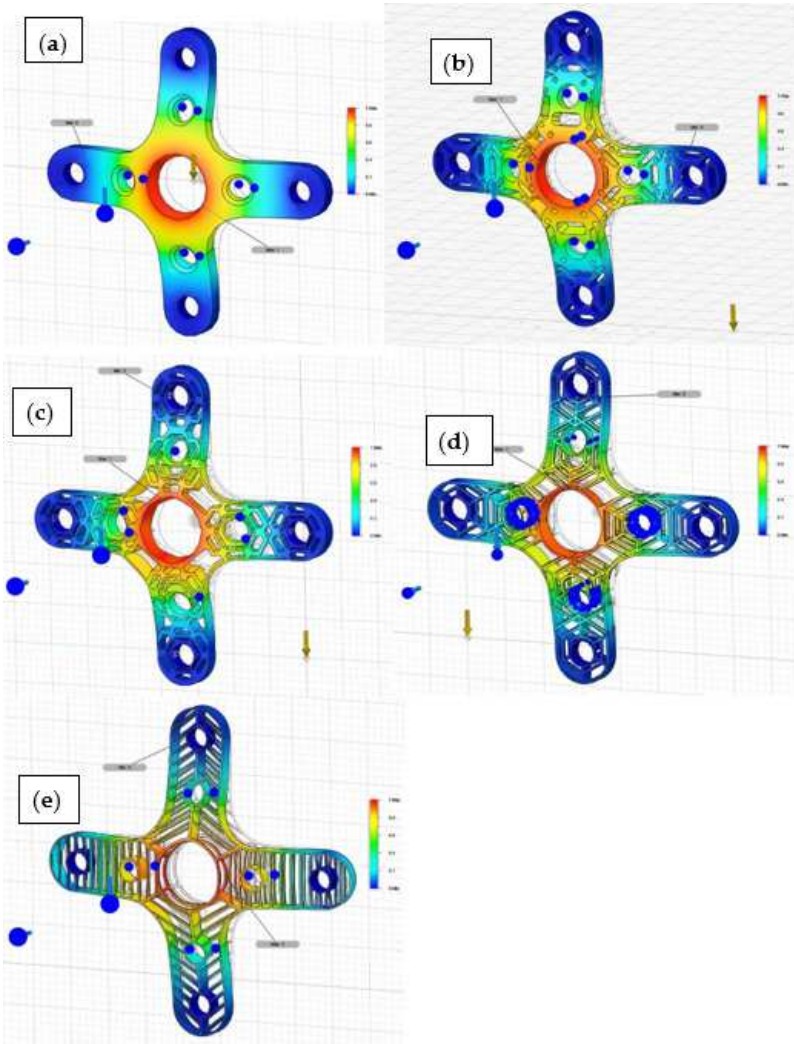

**Figure 13.** Modal frequencies simulation analysis of (**a**) original holder, (**b**) polyhedral, (**c**) snowflake, (**d**) circular spiderweb, and (**e**) linear spiderweb.

The value of mode 1 of the different holders must be higher than the vibration frequency due to the motor rotation, i.e., 166.66 Hz. The value of the different modes shows spiderweb has the lowest value to mitigate vibration of a brushless motor for each holder, as shown below. Table 7 shows all 8 frequency modes obtained from modal simulation. It clearly obvious that linear spider web has the loest Modal frequencies simulation.

**Table 7.** Modal frequencies simulation results.

|  | Original | Polyhedral | Snowflake | Circular Spider Web | Linear Spider Web |
|---|---|---|---|---|---|
| Mode 1 | 15,351 Hz | 9766 Hz | 10,530 Hz | 11,072 Hz | 9542 Hz |
| Mode 2 | 28,315 Hz | 19,081 Hz | 19,762 Hz | 17,756 Hz | 12,856 Hz |
| Mode 3 | 28,343 Hz | 20,078 Hz | 19,907 Hz | 21,895 Hz | 17,864 Hz |
| Mode 4 | 43,761 Hz | 33,069 Hz | 27,846 Hz | 21,945 Hz | 20,451 Hz |
| Mode 5 | 49,730 Hz | 34,996 Hz | 31,404 Hz | 26,289 Hz | 21,102 Hz |
| Mode 6 | 63,426 Hz | 35,249 Hz | 31,478 Hz | 28,467 Hz | 23,381 Hz |
| Mode 7 | 64,649 Hz | 38,256 Hz | 34,528 Hz | 30,596 Hz | 24,270 Hz |
| Mode 8 | 69,107 Hz | 39,823 Hz | 34,605 Hz | 30,987 Hz | 24,715 Hz |

All modes have a much higher value than the natural resonance of the motor, so they can absorb the vibrations of the motor. However, again linear spiderweb has the lowest value of frequency and hence less noise generation.

## 4. Discussion

Aircraft vibration is not something to be written off. If parts are not balanced correctly, the motor can risk cracking, failed avionics, and loss of engine performance. This can also contribute to metal fatigue which, if left unrepaired, can lead to potentially catastrophic engine failure. This is why monitoring aircraft vibrations is one of the most important aspects of aircraft maintenance. Therefore, we have investigated different holders to have the necessary characteristics to meet the customer's requirements, but some have better vibration damping properties than others. Comparing the frequency of mode 1, the holder with linear spider web unit cells has the highest noise reduction. It seems to be the most resistant to aerospace vibration. In the aeronautical field, the most important criterion is weight reduction. The lightest weight has the linear spider web holder with a mass of 0.3473 g and a porosity of 84%. The frequency of mode 1 is the lowest compared to the others tested and has an acceptable factor of safety.

The linear spider web holder fulfils the desired criteria as it minimises the weight while having good mechanical characteristics. Various parameters influence the mechanical characteristics of the holder, such as design, porosity, or even the chosen material. Although this study did not manufacture the part using selective laser melting (SLM), our major aim is to have a design concept for minimising the weight and reducing noise in brushless motors.

Our next step is to manufacture a linear spider web using our EOS 270M SLM facilities at the University of Wolverhampton under the supervision Dr. Klaudio Bari and sponsored by vertical aerospace Ltd. This technology is widely used in aeronautics since it allows the creation of complex parts and considerably lightens them. The use of structural lattice for aerospace parts allows us to reduce the quantity of material required and, therefore, less secondary waste and shorter manufacturing time. This represents an environmental and strategic target in this field since metals are becoming rarer. It would be interesting to carry out simulations with other materials and alloys, such as Ti64 and Maraging Steel, to see if the part can be further improved, but it is already noticeable that the mass of the initial holder has been reduced by 84% with the holder inspired by the spider's web.

## 5. Conclusions

This paper deals with conceptual additive layer manufacturing of vibration dampers for aerospace applications. The defense industry for aerospace is seeking a major improvement in fuel consumption and looking for green sources for powering their equipment. The results show a roadmap of how to mitigate vibration in aircraft using nature-inspired lattices. Spider web lattice structure resists stress in a stepwise fashion. After initially stiffening, the thread absorbs stress by stretching. Additional pressure causes the thread to sharply stiffen, thus transferring pressure to the rest of the web. This is a very similar strategy when the spider captures an insect in its web network. But even more, the pressure

is handled by a fourth and final process. Lattice structures within the spider web absorb the maximum vibration during a high spin of brushless motor. This resistance is similar to a struggling insect breaking into the spider web, but only those silk strands in contact with the insect can be affected. After a local thread or two break, the overall web strength increases. The study shows an ultimate load capacity increased by 20–30%, compared to its counterpart lattices. It is as though the web was designed to anticipate breaks. This study has shown that the support created from a spider's web fulfils the desired criteria and seems to be a good alternative to the original holder.

**Author Contributions:** Conceptualization, L.B. and K.B.; methodology, L.B.; software, L.B.; validation, L.B. and K.B.; formal analysis, K.B.; investigation, K.B.; resources, K.B.; data curation, K.B.; writing—original draft preparation, L.B.; writing—review and editing, K.B.; visualization, K.B.; supervision, K.B.; project administration, L.B.; funding acquisition, K.B. All authors have read and agreed to the published version of the manuscript.

**Funding:** This research was funded by European Regional Development (ERDF), under contract 32R19P03053 and the APC was funded by MDPI open access publishing in Basel/Switzerland.

**Acknowledgments:** We would like to acknowledge the support rendered by MDPI publishing and from Zwick Roell's Research team in Germany.

**Conflicts of Interest:** The authors declare no conflict of interest. Non-disclosure agreement has been signed between all parties in the project.

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
