# Peer review of "Spiderweb Cellular Structures Manufactured via Additive Layer Manufacturing for Aerospace Application"

_jcs, doi:10.3390/jcs6050133_

Round 1

Reviewer 1 Report

This is an interesting paper that covers a numerical simulation study for 3D printing with SLM. The paper reads well and flows nicely. I only have a couple of minor items:

  • Please remove the highlights in yellow.
  • The details on the FE models are quite superficial and light. Please ensure that your model is properly described to enable interested researchers from extending and replicating your work. Special attention should be paid to specifics such as element type (DOFs), convergence criteria and performance metrics.

  • Did the authors consider geometric imperfections? How about the effect of residual stresses?

Author Response

Reviewer comments

  1. The details on the FE models are quite superficial and light. Please ensure that your model is properly described to enable interested researchers from extending and replicating your work. Special attention should be paid to specifics such as element type (DOFs), convergence criteria and performance metrics.

Thanks for constructive feedback and I do agree that the FEA simulation in the previous version was not well explained, I have revised the whole manuscript and included vital information for the readers needed to replicate the modal frequency simulation, type of mesh (curvature) and detail of the boundary conditions. I have also included some figures and tables that help understanding why we are still in the design phase, and we were not ready for the manufacturing phase. So, if you compare the previous version with the revised version, you will notice a major revision around 90%.

  1. Did the authors consider geometric imperfections? How about the effect of residual stresses? This is very good point in ALM process, but, as mentioned we are still in the design phase, so we did not explore it yet. In term of stress concentration, so, we did not observe any concentrated stress that exceed 275/3 MPa which the safety margin elastic limit for aluminium.

Reviewer 2 Report

The manuscript deals with lattice structures for aerospace applications. Specifically, the research interest was aimed at structures, specifically designed to be used as vibration dampers for aerospace drone, to be produced by additive manufacturing (selective laser melting – SLM).

The authors compared four types of reticular bioinspired structures with the original support and validated them from a mechanical point of view through static and dynamic theoretical analyses, always considering weight minimization as the main criterion.

Overall, the work is interesting as it demonstrates further potential for the use of reticular structures for advanced applications in sectors for which lightening is a determining criterion and provides a further contribution to enhance the advantages of additive manufacturing. In any case, while appreciating the adequacy of the procedures adopted, before considering the manuscript to be definitively published, a careful re-reading of the paper is recommended to remove some typos, if only to make the reading of the text more fluent. For example:

Paragraph 1 "Introduction" - last line: it is advisable to change the text "cyclic, fatigue, testing and low speed impact testing" to "cyclic, fatigue and low speed impact testing".

Author Response

Thanks for constructive feedback and I do agree that an improvement in written English languages were needed, I have revised the whole manuscript and included vital information for the readers willing  to replicate the modal frequency simulation, type of mesh (curvature) and detail of the boundary condition. I have also included some figures and tables to help understand why we are still in the design phase, and we were not ready for the manufacturing phase.

So, if you compare the previous version and the current version, you will notice a major revision around 90%.